# Study of Gas Film Characteristics in Electrochemical Discharge Machining and Their Effects on Discharge Energy Distribution

**DOI:** 10.3390/mi14051079

**Published:** 2023-05-20

**Authors:** Hao Liu, Adayi Xieeryazidan

**Affiliations:** School of Mechanical Engineering, Xinjiang University, Urumqi 830017, China; lh735568216@163.com

**Keywords:** ECDM, gas film characterization, discharge energy distribution, machining quality

## Abstract

Glass is a hard and brittle insulating material that is widely used in optics, biomedicine, and microelectromechanical systems. The electrochemical discharge process, which involves an effective microfabrication technology for insulating hard and brittle materials, can be used to perform effective microstructural processing on glass. The gas film is the most important medium in this process, and its quality is an important factor in the formation of good surface microstructures. This study focuses on the gas film properties and their influence on the discharge energy distribution. In this study, a complete factorial design of experiments (DOE) was used, with three factors and three levels of voltage, duty cycle, and frequency as the influencing factors and gas film thickness as the response for the experimental study, to obtain the best combination of process parameters that would result in the best gas film quality. In addition, experiments and simulations of microhole processing on two types of glass, quartz glass and K9 optical glass, were conducted for the first time to characterize the discharge energy distribution of the gas film based on the radial overcut, depth-to-diameter ratio, and roundness error, and to analyze the gas film characteristics and their effects on the discharge energy distribution. The experimental results demonstrated the optimal combination of process parameters, at a voltage of 50 V, a frequency of 20 kHz and a duty cycle of 80%, that achieved a better gas film quality and a more uniform discharge energy distribution. A thin and stable gas film with a thickness of 189 μm was obtained with the optimal combination of parameters, which was 149 μm less than the extreme combination of parameters (60 V, 25 kHz, 60%). These studies resulted in an 81 μm reduction in radial overcut, a roundness error reduced by 14, and a 49% increase in the depth–shallow ratio for a microhole machined on quartz glass.

## 1. Introduction

In recent years, owing to their essential properties, such as optical transparency, chemical stability, high-temperature resistance, and insulation properties, brittle glass materials have been widely used for manufacturing microproducts [1,2], such as solid oxide fuel cells, DNA arrays, microreactors, micropumps, and flow sensors [3,4]. However, as glass is a hard and brittle insulating material, machining is extremely challenging using traditional methods. Therefore, some nontraditional processing methods have been used, such as ultrasonic machining, laser beam machining, and abrasive flow machining. Nonetheless, nontraditional machining methods are ineffective in resolving problems such as microcracks in glass micromachining, thereby limiting their widespread use in this field [5,6,7,8,9,10]. In a previous study, a new composite machining method, electrochemical discharge machining (ECDM), was proposed to improve the processing quality. ECDM was developed based on the electrode discharge effect during electrochemical machining [11]. Kurafuji introduced ECDM in 1946 and applied it to glass drilling [12].

As the primary medium generated by the discharge in ECDM, the gas film quality will significantly affect the machined etch pits or microhole quality. Therefore, the study of the characteristics and quality of the gas film becomes especially crucial. In 2004, R. Wüthrich and H. Bleuler investigated the theoretical model and the critical conditions required to form a gas film, which provided the theoretical foundation for further investigations on gas film characterization using ECDM [13]. Subsequently, R Wüthrich and Hof et al. found that gas film thickness is the main limiting factor for the occurrence of electrochemical discharges and that gas film thickness is influenced by the surface tension of the electrolyte [14]. In 2010, Cheng et al. reported the characterization of gas films by profile shape and size and studied the importance of gas film stability on discharge stability. Moreover, the data obtained from the discharge current signal revealed changes in the structure of the gas film, thus providing a valuable reference for subsequent studies on changing the process parameters to enhance machining efficiency and accuracy [15]. In addition, Kolhekar and Sundaram validated a technique for characterizing gas films using discharge current data by applying a complete factorial parameter method [16]. They used the standard deviation of the discharge energy and discharge current data to predict gas film thickness and gas film stability, respectively. Singh experimentally investigated the hydrogen gas film thickness during a triplex hybrid process of rotary mode (RM) ECDM. The effect of the gas film thickness on the energy channelization behavior on the RM-ECDM process improved the processing quality [17]. Huang et al. investigated gas film breakdown characteristics under different power supply parameters and further analyzed the relationship between process parameters and the formation of the quality and material removal rate under gas film stability conditions [18]. To process a variety of arbitrary shapes, Appalanaidu proposed a method for controlling the shape of the air film by controlling the electrostatic force acting on the bubbles [19]. Subsequently, Liu et al. discovered the phenomenon of chain discharge caused by the expansion and morphology of the gas film for the first time. Chain discharge as a new phenomenon can help us to understand the mechanism and analyze the processing results of ECDM [20]. By observing the discharge effect on the gas film, Tang et al. found that the shape of the gas film is influenced by the forces acting during the discharge process. In addition, the energies released by different types of discharge were calculated according to the voltage and current waveforms [21].

This study demonstrated that gas film, which is the most crucial medium for changing the machining state from electrolysis to EDM in ECDM, significantly impacts the precision and quality of machining. The impact of power supply parameters, particularly their interaction, on electrochemical discharge processing has not been adequately investigated in the past. Furthermore, the characteristics of the gas film and its influence on the discharge energy distribution need to be further studied. Therefore, this study used three-factor, three-level complete factorial experiments to optimize the process parameters and investigate the effects of various power supply parameters on the gas film characteristics. In this study, the thickness of the gas film was considered the primary factor, and its stability was the secondary factor in describing its quality. In addition, we also investigated the effects of gas film properties on the discharge energy distribution during ECDM.

## 2. Experimental Method and System

### 2.1. Experimental Setup

Figure 1 shows the ECDM experimental platform, which consists of an ECDM device and an experimental observation device. The ECDM device consisted of a three-coordinate control platform and computer control system, a DC pulse power supply, an electrolytic bath on a worktable, an electrolyte, an anode (auxiliary electrode), and a cathode (tool electrode). The minimum feed accuracy of the three-coordinate control stage is 1 μm. The observation device comprised a high-speed camera (alhua A5131MU210, Dahua Technology, Hangzhou, China) and oscilloscope (Tektronix TBS102X Digital Storage Oscilloscopes, Tektronix Corporation, Johnstown, OH, USA). The electrolytic bath was made of transparent plexiglas, which allowed the high-speed camera to capture the gas film image and the entire electrochemical discharge process in real-time. The high-speed camera exposure was adjusted to 100 to capture a clear image of the gas film. In contrast, the interval between two adjacent image acquisitions was set to 100 ms according to the experimental requirements. The oscilloscope recorded the current and voltage signals during the discharge process. The VHX-6000 Ultra-Depth Microscope (Keenes, Osaka, Japan) was used for the observation and data acquisition of microbores that were machined.

### 2.2. Materials and Methods

The experiments used a DC pulse power supply as the processing power source. The plexiglas electrolytic bath was a rectangular tank with 12 × 10 × 7 cm${}^{3}$ and a wall thickness of 1 cm. The auxiliary anode was made of graphite, and a graphite plate with dimensions of 50 × 40 × 4 mm${}^{3}$ was selected. The tool electrode was made of tungsten carbide, and columnar electrodes with dimensions of 150 mm and 300 μm were used. The electrolyte was selected from 1 mol/L sodium hydroxide solution. The workpiece materials selected were 1 mm thick optical and quartz glasses. The selected experimental materials and parameters for ECDM are presented in Table 1.

The test was conducted using a three-factor, three-level complete factorial DOE, with voltage, frequency, and duty cycle as the input parameters, as shown in Table 2. The other test parameters included the distance between the electrodes and the electrolyte concentration and level. Based on the reviewed literature, a large electrode spacing, low electrolyte concentration, and high electrolyte level were applied in this study, as shown in Table 1. The experimental images were acquired using a high-speed camera and oscilloscope, and the data extraction and analysis of the acquired gas film images were performed using Image J (1.8.0) and Minitab 19 (19.1.0.1).

Figure 2 illustrates the electrochemical gas film image captured from the discharge machining and processing system. The experiment was conducted in two stages: the first stage involved air film generation and image acquisition, and the second stage involved the experimental process. In the first stage of the test, the tool electrode was clamped on the z-axis of the three-coordinate platform and partially submerged in a NaOH electrolyte with the auxiliary anode. To observe the entire gas film formation process better, we adjusted the buffer time of the DC pulse power supply to 50. After turning on the power supply, the critical voltage is often reached in the eighth second when the gas film breaks to produce a discharge spark. A high-speed camera acquired images of the gas film formation process. The image created before the gas film breakdown was captured to represent the gas film characteristics accurately. Furthermore, the edges of the gas film were extracted using MATLAB edge detection software. The edges were also traced from 25 different points to confirm data accuracy, and the diameter of the resulting picture was estimated using the ImageJ measuring program, as shown in Figure 3b. Subsequently, the tool electrode diameter was deduced from the measurement data. Next, the obtained data were segregated into two parts to determine the thickness of the gas film. The Equation (Equation 1) for calculating the thickness of the gas film is given by:(1)$$H=[\frac{\left(d(1)+d(2)+\cdots +d(n)\right)}{n}-D]/2$$

In the above equation, *H* is the total gas film thickness, d(n) is the gas film thickness at the nth position, *n* is the number of measurement positions, and *D* is the diameter of the tool electrode. The measured gas film thickness data were imported into Minitab 19. Using a complete factorial analysis, the reliability of the data was analyzed, and the test results were derived. In the second stage of the experiment, two types of glass were processed, and the final experimental results were verified through simulations.

## 3. Results and Discussion

### 3.1. Gas Film Formation

ECDM involves two stages of processes: electrochemical and discharge. The gas film acts as a breakdown medium throughout the ECDM process, thereby allowing the machining state to change from electrochemical to discharged. The entire process of gas film formation was observed more distinctly with an increase in voltage when the buffer value of the DC pulse power supply was adjusted to 50. As shown in Figure 3a, at the beginning of energization, no gas film was observed because of the small voltage. During the experiment, more bubbles gathered on the tool electrode and formed a dense and stable gas film as the voltage increased.

Figure 4a illustrates the voltage–current characteristic curve of the ECDM process. The Figure 4b blue area indicates the process of one discharge generation in the ECDM from the formation of the gas film to its breakdown. The picture indicated by the arrow shows its voltage–current characteristic curve. The gas film was formed by the accumulation of the hydrogen bubbles formed by electrochemical reactions and the vapor bubbles formed by electrolyte evaporation. As shown in Figure 5, the ECDM process can be divided into five stages. Using these images, the process of gas film formation can be analyzed visually and more scientifically. The OA section is thermodynamic and occurs over the potential region; at this stage, owing to the low voltage, a current is not generated, the electrochemical reaction does not begin, and bubbles are not formed. The AB section is a linear zone: the relationship between the voltage and current at this stage obeys Ohm’s law, and the current increases linearly with increasing input voltage. Furthermore, an electrochemical reaction begins, a pathway is formed between the two electrodes, and a few hydrogen bubbles are formed on the surface of the tool electrode. BC section is the saturation zone: from point B onwards, the bubbles generated on the surface of the tool electrodes increase and begin to gather and merge. The current value with voltage trend progressively stabilizes at this point. However, the gas layer is not sufficiently dense or stable, which causes an increase in the resistance between the two electrodes. Section CD is the unstable region; as shown in Figure 5, the current is significantly reduced as the voltage increases. Numerous hydrogen bubbles generate, gather, and merge on the surface of the tool electrode, thereby causing the gas film to form a dense and stable structure. At this stage, a dense and stable insulating gas film is generated, the current gradually converges to zero, a potential difference is formed inside and outside the gas film, and some tiny unstable sparks are generated. Simultaneously, many oxygen bubbles are generated on the surface of the auxiliary anode; however, their large surface areas do not allow bubbles to adhere to the air film. The DE section is the discharge area: at this stage, the gas film formed is dense and stable, and, as the input voltage continues to increase, the potential difference inside and outside of the gas film increases. When the potential difference exceeds the critical value, the gas film is broken, thereby resulting in an electrochemical discharge. The actual process of electrochemical discharge processing occurs in this stage. The gas film breaks to produce a spark discharge.

### 3.2. Gas Film Thickness

Table 3 presents the analysis of variance (ANOVA) for the air film thickness. The F-values and *p*-values are shown in the table to determine the significance of the influencing factors. Factors with large F-values (*p* < 0.05) were considered significant. Table 3 shows that the voltage and duty cycle interaction had a large F-value and a *p*-value = 0. The interaction between frequency and duty cycle had a *p*-value = 0.01. This indicates that the interaction significantly affected the thickness of the gas film.

The plots of the main effect of the average gas film thickness are shown in Figure 6. The plots show that the impact of voltage and duty cycle on the gas film thickness was significant, whereas the effect of the frequency alone on the gas film thickness was insignificant. The central effect diagram shows that a minimum voltage, maximum duty cycle, and frequency are the best technological parameters for forming a thin gas film.

As shown in Figure 6a, the thickness of the gas film increased as the voltage increased. Firstly, as the voltage increased, the electrochemical reaction became more violent, thus resulting in more bubbles per unit of time. The bubble accumulation rate became faster as the voltage increased and the gas film thickened. Secondly, as the voltage increased, the electrochemical discharge became more intense. A large amount of heat generation accompanied this process, and the electrolyte was evaporated, thus producing a large amount of gas. These two factors are the main reasons for the excessive thickness of the gas film as the voltage increased. As shown in Figure 6b, the effect of frequency on film thickness was minimal, with a minimal reduction in film thickness as the frequency increased. This is because the increase in frequency resulted in a shorter turn-on time per unit time, thus resulting in fewer bubbles being generated with less reaction time per turn-on, which had a definite effect on the reduction in film thickness. Still, the overall change in reaction time was small and, therefore, did not significantly affect the film thickness. As shown in Figure 6c, the duty cycle also had a more significant effect on the film thickness; as the duty cycle reached a maximum (80%), the proportion of time spent on power during a pulse decreased, which may have prevented the consolidation of large bubbles generated by excessive voltage and reduced the film thickness. Therefore, when the interaction effects were not considered, the best experimental parameters for the air film thickness were a voltage of 50 V, a frequency of 30 kHz, and a duty cycle of 80%. The extreme experimental parameters for air film thickness were a 60 V voltage, 20 kHz frequency, and 60% duty cycle. In both cases, the gas film thicknesses were 214 μm and 335 μm, respectively.

The ANOVA in Table 3 shows that the effects of the independent factors mentioned above did not provide a more comprehensive indication of the influence of these factors on the gas film characteristics. Therefore, the effect of the two-factor interaction was considered, as shown in Table 3. The interactions of the voltage–duty cycle and the frequency–duty cycle significantly affected the thickness of the gas film. Figure 7 shows the plots of the interaction effects of voltage and duty cycle, as well as frequency and duty cycle, on the thickness of the gas film. The interaction of voltage and duty cycle in Figure 7a shows that, at 60% duty cycle, the film thickness decreased linearly with decreasing voltage; at 70% duty cycle, the film thickness also decreased with decreasing voltage, but the effect was not significant; at 80% duty cycle, a small voltage was more favorable for thin film formation. Generally, the film thickness decreased with increasing voltage for a constant duty cycle. Still, since a small duty cycle leads to a longer pulsed power supply charging process, the film thickness decreased with increasing voltage. However, since a small duty cycle results in a longer charging process of the pulsed power supply, the number of bubbles generated increased, thus increasing the film thickness. Therefore, to obtain a thin gas film, a combination of low voltage and a high combination of low voltage and the high duty cycle was an ideal choice to obtain a thinner gas film. The interaction of frequency and duty cycle in Figure 7b shows that, when the duty cycle was greater than 70%, the film thickness decreased as the frequency decreased. At a constant frequency, it is also clear that the film thickness decreased as the duty cycle decreased. It is also clear that the film thickness decreases as the duty cycle increases. Hence, the combination of maximum duty cycle and minimum frequency provides favorable conditions for forming thin films. This may be because the power supply has less time during a high-duty cycle. This may be because the power supply has less energization time in a high-duty cycle, thus resulting in less bubble formation and lower film thickness.

In summary, voltage, duty cycle, their interaction, and the interaction of frequency and duty cycle are the factors that significantly affect the gas film thickness. By combining the analyses of the effects of the single-factor and two-factor interactions on the thickness of the gas film, the response optimization of the data determined the values of minimum voltage, minimum frequency, and maximum duty cycle and the optimal combination of these process parameters for forming a thin gas film. The optimal process parameter combination was 50 V, 20 kHz, 80% duty cycle, and the extreme process parameter combination was 60 V, 25 kHz, and 60% duty cycle. Their gas film thicknesses were 189 μm and 338 μm, respectively. Figure 8 compares the thickness of the gas film for the extreme experimental and optimal process parameters.

### 3.3. Gas Film Stability

In this study, the gas film stability was considered a secondary factor in determining the gas film quality. The ability of a gas film to maintain its shape during the discharge process is referred to as gas film stability. A stable gas film was required to ensure the strength of the discharge during the experiment. Therefore, the standard deviation of the electric discharge current values was used as a predictive factor for analyzing the stability of the gas film shape to estimate whether the ideal combination of process variables for producing a thin gas film resulted in the simultaneous production of a more stable gas film. The standard deviations of the discharge current values were calculated to be 0.21 and 0.25 for the optimal and extreme combinations of process variables, respectively. A lower standard deviation corresponds to higher stability of the gas film [15], and the optimal combination of process parameters was found to be the best condition for generating thin and more stable gas films in ECDM. This verifies that the optimum combination of process variables obtained by our experimental analysis could lead to better gas film quality in ECDM.

## 4. Experimental Validation and Simulation

Since electrochemical discharge differs from other EDM methods in the form of discharge (spark discharge by breaking the gas film), this paper proposed an ECDM discharge energy distribution to provide new ideas for the microscopic quality of ECDM and establish a simple relationship between it and machining quality. The experimental results showed that, when an optimal combination of process parameters was used in ECDM, the generated gas film was thinner and more stable in shape, thereby implying that the discharge was more stable and required less energy to break down the gas film. The machining results reflected this, because the microholes produced with the optimal combination of process parameters had better surface topography and machining accuracy. In contrast, in ECDM, the thickness and stability of the gas film formed experimentally using a combination of extreme process parameters were poor. In addition, the differences in the microscopic morphology of the machined microholes reflected the differences in material removal at different locations owing to an uneven discharge energy distribution. The experimental results were verified by comparing the experimental and simulation parameters listed in Table 4.

In this study, the processed micropores depths of less than 300 μm. Studies have shown that thermal removal in the ECDM is the primary material removal method for holes less than 300 μm deep. Therefore, COMSOL multiphysics software was used to simulate the ECDM’s material removal process. The simulation was performed under the ideal condition of a single spark to obtain the material removal process for ECDM. In Figure 9, the top half of the image shows the simulated material removal. The isotherms in this section of the image show that the temperature distribution on the inner walls of the microholes was more uniform, and the isotherms were smoother for the best process parameters. This is due to the fact that the gas film generated under optimal process parameters was stable and had a relatively uniform thickness distribution at all locations, thus resulting in a uniform and stable distribution of the discharge energy generated during the breakdown of the gas film. The figure’s bottom half shows the material removed during the experimental process. The images show that the maximum depth of material removal simulated at the optimal process parameters was 119 μm, which constitutedan increase of 23 μm from the depth observed using the extreme process parameters. Moreover, the experimental data in the figure verify that the optimal combination of process parameters resulted in higher material removal rates and greater hole depths. This is because the discharge energy generated in the ECDM was more concentrated at the bottom of the gas film when the process variables were optimized. Consequently, the material was removed longitudinally, and the microholes were deeper.

During this experiment, the two types of glasses were processed for 100 s using a feed rate of 1.5 m/s. The radial overcut (pa) of the inlet, roundness (f), and the depth-to-diameter ratio of the hole (L/D) were calculated from the experimental results and are listed in Table 5. The experimental results show that minor radial overcuts, smaller roundness errors, and larger depth-to-diameter ratios were obtained for the microholes processed on both glasses with the optimal combination of process parameters. The entrance shapes and radial overcuts of the machined microholes are shown in Figure 10. The study results in Figure 10 and Table 5 demonstrate a significant reduction in the number of microholes machined on both glasses towards radial overcut for the optimum combination of process parameters. This was associated with the difference in the discharge energy distribution caused by gas film differences. Because the generated gas film was thinner and more stable in ECDM with the optimal combination of process parameters, the energy required to break down the gas film was low, and the film was more stable during discharge. In addition, the discharge was more concentrated at the bottom of the electrode, and the sidewall discharge was lower than the extreme process parameters, which is the main reason for the lower radial overcut.

The experimental studies have shown that energy distribution affects microhole roundness and roughness more. The results in Table 5 and Figure 11 reveal that a continuous and stable discharge was generated in the ECDM process owing to the generation of stable gas films with an optimal combination of process parameters. Using extreme process parameters in ECDM leads to a large gas film thickness and poor stability, thus resulting in an uneven distribution at various locations in the gas film. For example, the generated discharge energy was small at sites with large gas film thicknesses, thus resulting in irregular bumps due to insufficient material removal. A thinner gas film produces more discharge energy and results in excessive material removal.

According to the experimental results, the energy required to break through the gas film was lower at locations with lower thicknesses. The energy consumed by this process was relatively less. Therefore, the energy generated by the discharge after breaking through the gas film was one of the reasons for uneven material removal. This was reflected in the microhole morphology, which had a more significant roundness error. Furthermore, Figure 11a shows that the surface of the ECDM-machined microhole was smooth when the discharge energy was evenly distributed, and the entrance edge was more regular than the right edge and without too many concaves- or convex-shaped irregular surfaces.

According to the study’s findings, with the optimal combination of process parameters, a thin and stable gas film is formed in the ECDM process; thus, the discharge energy generated by the electrochemical discharge is more uniformly distributed in all positions. Moreover, it was found that a reduction in the gas film thickness could also decrease the gas film stability. A thinner gas film breakdown was associated with lower discharge energy, which caused less turbulence in the surrounding electrolyte and poorer gas film stability. Differential discharge energy distribution caused by changing gas films affected the quality of micro-hole machining and surface micro-forming. The concentration of the discharge energy distribution resulted in a high material removal rate, while the uniformity of the discharge energy distribution largely determined the surface microforming quality of the processed microstructure. This is reflected in the fact that the depth of the microhole was more profound when the discharge energy was concentrated at the bottom of the tool electrode, the roundness error of the microhole was more minor, and the roughness was lower when the discharge energy was uniformly distributed. Combined with the analysis of the experimental results, we conclude the following:(1)The influence of air film on the discharge energy distribution: the difference between the thinness and thickness of the air film determines the different discharge energy distributions on the air film. The discharge energy distribution is low where the air film thickness is large and high where the air film thickness is thin;(2)The influence of discharge energy distribution on surface micromorphology and processing quality: the impact of discharge energy distribution on surface micromorphology is the most significant. The location of high energy generation has a high material removal rate, and the site of low energy has less material removal, so the uniform degree of discharge energy distribution in each location on the workpiece surface determines the surface roughness and roundness of the microhole to a certain extent, and reduces the entrance edge to a certain extent. Secondly, the concentrated distribution of discharge energy at the bottom of the air film can effectively improve the depth of the microhole, and the reduced distribution at the side walls of the air film can also effectively minimize radial overcutting and improve the depth-to-diameter ratio.

## 5. Conclusions

The study analyzed gas film characteristics during ECDM using a three-factor, three-level complete factorial DOE. The study used power supply parameters as variables. Gas film thickness was used as the primary quality measure, and gas film stability was investigated as a secondary quality measure. The optimum process parameter combination for minimum film thickness and strength was identified. Moreover, the discharge energy distribution of ECDM was presented, and the effect of varying the gas film properties on the discharge energy distribution was analyzed. Heat removal simulations and experimental machining were carried out. Silica glass and optical glass were used as machined workpieces. The main conclusions are the following:(1)The optimum combination of process parameters for the generation of thin and stable gas films was found through response optimization (voltage 50 V, frequency 20 kHz, and 80% duty cycle); the experimental results showed that the thickness of the gas film was reduced by 149 μm and the stability of the gas film was improved to some extent compared with the gas film generated by the extreme combination of process parameters;(2)A simple relationship between the discharge energy distribution and the machining quality and surface microstructure of the microhole was established; that is, the location distribution of the discharge energy was determined by the thickness of the gas film at each location, and the discharge energy distribution significantly affected the surface microstructure of the microhole. The degree of uniformity of the energy distribution determines the surface roughness and roundness of the microhole to a certain extent and affects the radial overcut and hole depth. This provides a feasible solution and a new direction for the improvement of the micromorphology of the surface of the microhole in electrochemical discharge machining;(3)The processing quality was also greatly affected by the improvement in the quality of the gas film. The experimental result shows that the microhole depth on the quartz glass was reduced by 81 μm, the depth-to-diameter ratio was improved by 49%, and the roundness error was reduced by 14.

## Figures and Tables

**Figure 1 micromachines-14-01079-f001:**
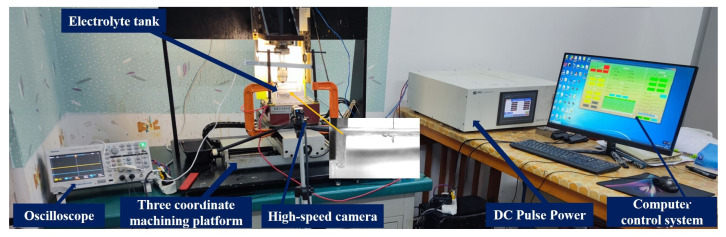
ECDM experimental platform.

**Figure 2 micromachines-14-01079-f002:**
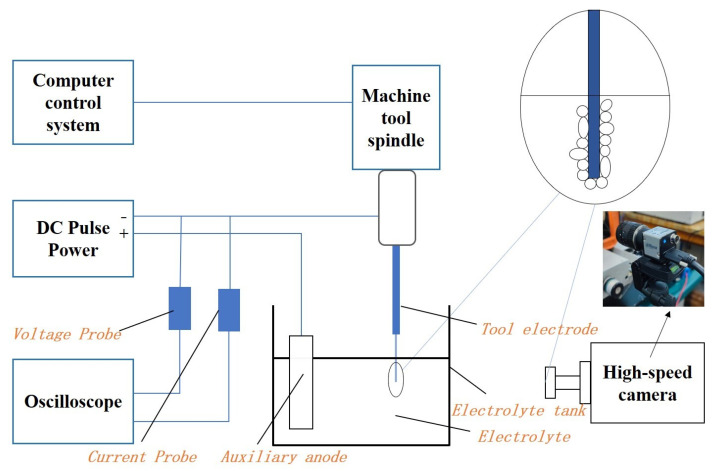
Schematic diagram of electrochemical discharge processing gas film image capture and processing system.

**Figure 3 micromachines-14-01079-f003:**
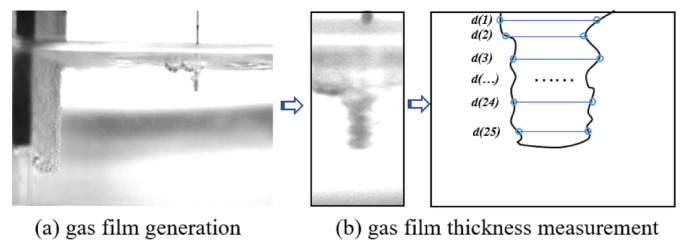
Gas film generation and thickness measurement diagram.

**Figure 4 micromachines-14-01079-f004:**
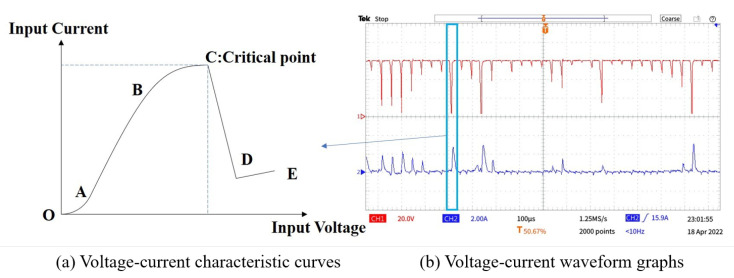
Voltage–current characteristic curve of electrochemical discharge machining process.

**Figure 5 micromachines-14-01079-f005:**
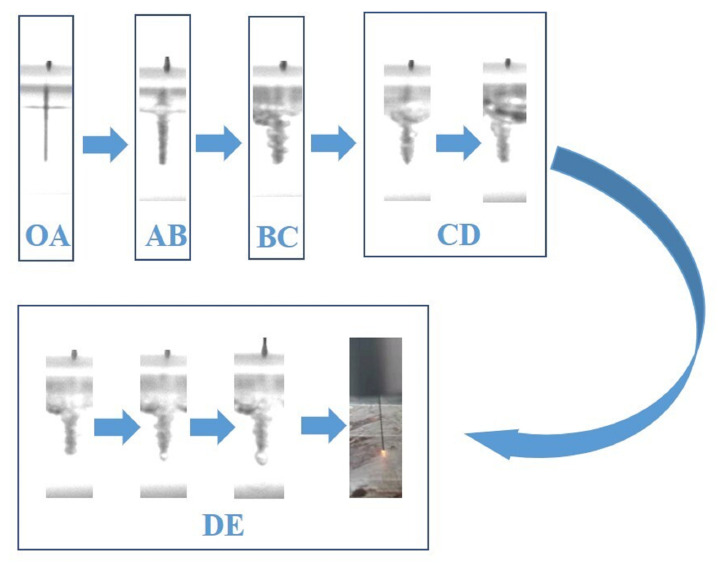
Electrochemical discharge machining process.

**Figure 6 micromachines-14-01079-f006:**
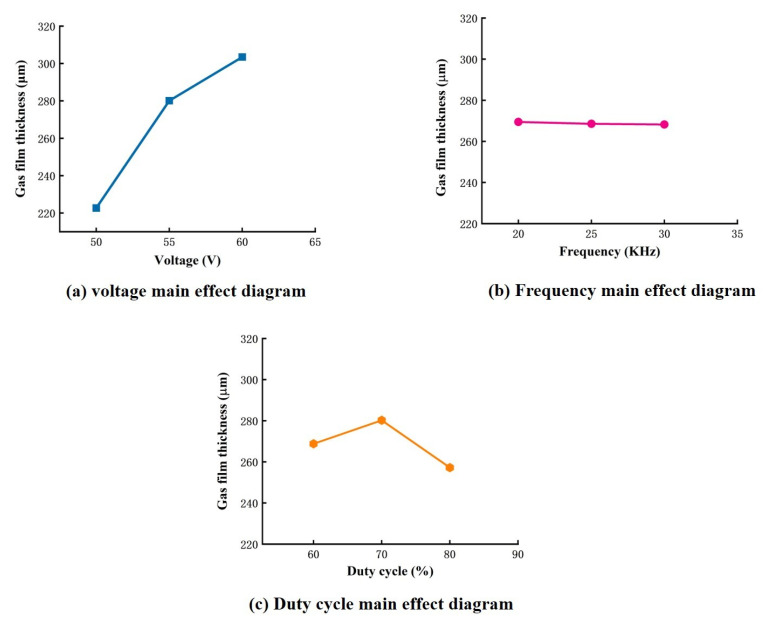
Main effect diagrams of the mean value of gas film thickness.

**Figure 7 micromachines-14-01079-f007:**
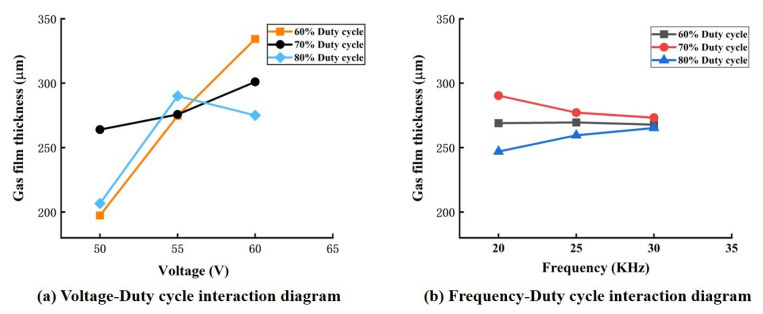
Gas film thickness two-way interaction effect diagram.

**Figure 8 micromachines-14-01079-f008:**
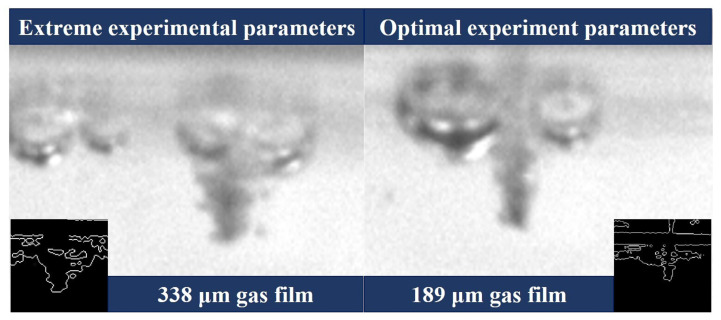
Gas film thickness at extreme and optimal process parameters (inset shows the edge detection image for calculating gas film thickness).

**Figure 9 micromachines-14-01079-f009:**
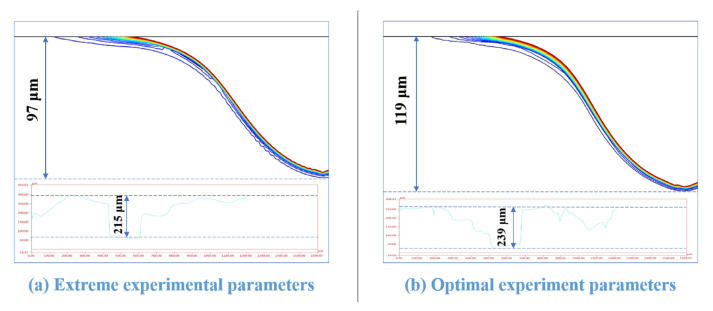
Electrochemical discharge machining material removal COMSOL simulation and machining results (inset shows actual machining depth).

**Figure 10 micromachines-14-01079-f010:**
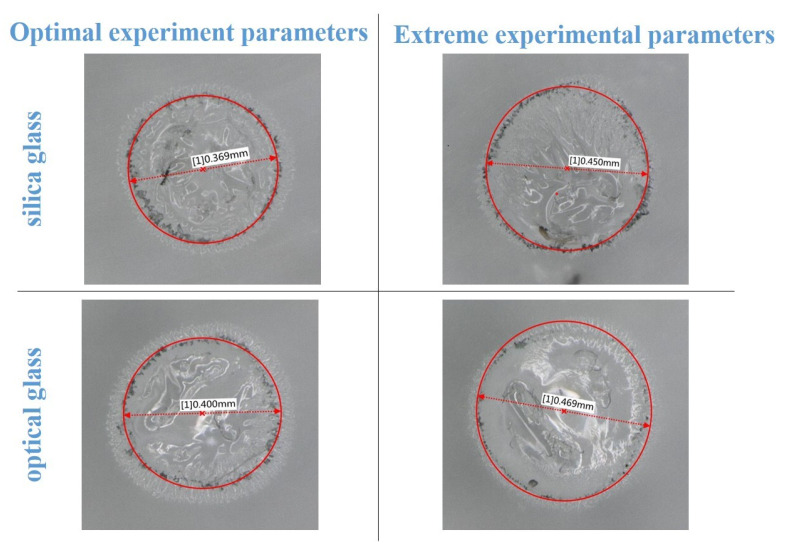
Micropores produced on two types of glass using different process parameters.

**Figure 11 micromachines-14-01079-f011:**
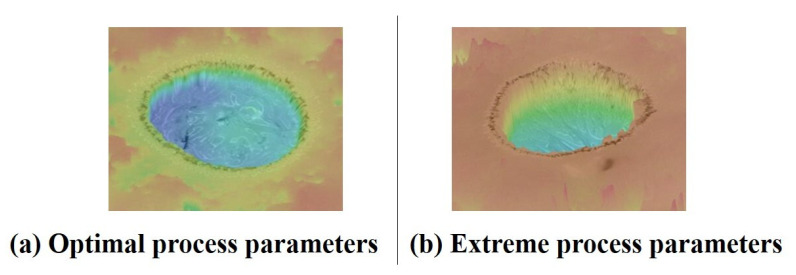
Surface morphology of microhole by electrochemical discharge machining.

**Table 1 micromachines-14-01079-t001:** Electrochemical discharge machining of each experimental device and material parameters.

Name	Parameters
Workpiece	Silica glass/Optical Glass: h = 1 mm
Tool electrode	Tungsten carbide: D = 300 μm
Auxiliary anode	Graphite: 50 × 40 × 4 mm${}^{3}$
Electrode spacing	3.5 cm
Electrolyte	NaOH
Electrolyte concentration	1 mol/L
Electrolyte level	5 mm

**Table 2 micromachines-14-01079-t002:** Process parameters considering level changes.

Factors	Level 1	Level 2	Level 3
A: Voltage (V)	50	55	60
B: Frequency (kHz)	20	25	30
C: Duty cycle (%)	60	70	80

**Table 3 micromachines-14-01079-t003:** Analysis of variance of gas film thickness.

Source	DF	Adj SS	Adj MS	F-Value	*p*-Value
Model	18	45,985.8	2554.8	71.11	0.000
Linear	6	33,495.8	5582.6	155.39	0.000
A	2	31,108.1	15,554.0	432.95	0.000
B	2	7.2	3.6	0.10	0.906
C	2	2380.5	1190.3	33.13	0.000
2-way Interaction	12	12,490.0	1040.8	28.97	0.000
A*B	4	295.9	74.0	2.06	0.179
A*C	4	11,187.3	2796.8	77.85	0.000
B*C	4	1006.8	251.7	7.01	0.010
Error	8	287.4	35.9		
Total	26	46,273.2			

**Table 4 micromachines-14-01079-t004:** Determined combination of process parameters.

Factors	Optimal	Extreme
A: Voltage (V)	50	60
B: Frequency (KHz)	20	25
C: Duty cycle (%)	80	60

**Table 5 micromachines-14-01079-t005:** Summary of experimental results.

Factors	Silica Glass		Optical Glass	
	Optimal	Extreme	Optimal	Extreme
pa	69 μm	150 μm	100 μm	169 μm
f	7	21	3	17.5
L/d	0.596	0.400	0.598	0.458

## Data Availability

Not applicable.

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
