# Peer review of "Study of Gas Film Characteristics in Electrochemical Discharge Machining and Their Effects on Discharge Energy Distribution"

_micromachines, 2023, doi:10.3390/mi14051079_

Round 1

Reviewer 1 Report

Dear author,

The paper "Study of gas film characteristics in electrochemical discharge machining and its effect on discharge energy distribution" is good and logical. And also, it is significant to focuses on the gas film properties and their influence on the discharge energy distribution. This paper has been reviewed but it needs major revision before accepted. The followings are the points need to modify.

1.    In abstract, it should have some results of the paper.

2.    The Introduction is too long. It shall be shortened and essential.

3.    The reference should be cited like this [1-2], not this [1][2].

4.    Table is not an abbreviation, so there is no need to have dot after it.

5.    In equation (1), there should have “…” between d(2) and d(n)

6.    Fig. 4 should represent figure 4(a) 4(b) separately and added description of it. So do other figure.

7.    In table 3, how do you get adj ss in A*B and A*C? the author should have some discussion able table 3.

8.    In line 270, please note that there should have a blank after Fig. 10. The author should check the grammar.

9.    In line 293, the figure should be figure 11(a) and (b), not left or right

10. The conclusion need to be more specific and more detail results. It is suggested that put the results point by point.

11. Also, the author should cite the lately reference, most of the references are long ago. There should be more new paper about the topic.

please see "Comments and Suggestions for Authors
"

Reviewer 2 Report

This paper focuses on studying the gas film properties and their influence on the discharge energy distribution. The research was undertaken using scholarly approaches, it would be more useful to readers if following  modification can be made: 

1、The novelty of the work is unclear. There are many publications about gas film in ECDM process.Study of gas film quality in electrochemical discharge machining, International Journal of Machine Tools & Manufacture 50 (2010) 689–697). The novelty needs to be identified and highlighted in the contribution.

2、A machining setup is shown in Fig. 1. There is no reference on the set-up and related instrument/component whether it is home built or commercially purchased. If purchased, then what are the brand names? If homebuilt, then what are the specifications and accuracy? Such info is necessary in a quality research paper and would give credibility to the data and assure readers of the accuracy and repeatability of data.

3、Only three points were studied for each parameter, so the experiments of this study are not enough.

4、The paper only describes the experiment results, but it lacks of the theoretical analysis of the machining mechanism.

5、Many of the figs are not clear. It's difficult to distinguish between a gas layer or an accumulation of bubbles (fig. 8).

6、The format of the reference should be modified according to formal format.

English is understandable.

Round 2

Reviewer 1 Report

Dear author,

This modified paper is fine. There is only one thing need to recheck. All the figure should in the same presenting style. For example, figure 3. The a and b is written in the illustrations of the figure (after it). But figure 4 is under the figure. Please unity the style

N/A

Reviewer 2 Report

minor revision: fig.5 and fig.8 are not clear.
